# Research on Electro-Optical Characteristics of Infrared Detectors with HgCdTe Operating at Room Temperature

**DOI:** 10.3390/s23031088

**Published:** 2023-01-17

**Authors:** Paweł Madejczyk, Tetiana Manyk, Jarosław Rutkowski

**Affiliations:** Institute of Applied Physics, Military University of Technology, 2 Kaliskiego St., 00-908 Warsaw, Poland

**Keywords:** HgCdTe heterostructure, infrared detectors, HOT, current–voltage characteristics

## Abstract

This paper presents a thorough analysis of the current–voltage characteristics of uncooled HgCdTe detectors optimized for different spectral ranges. HgCdTe heterostructures were grown by means of metal–organic chemical vapor deposition (MOCVD) on GaAs substrates. The obtained detector structures were measured using a Keysight B1500A semiconductor device analyser controlled via LabVIEW for automation. The experimental characteristics were compared with numerical calculations performed using the commercial platform SimuAPSYS (Crosslight). SimuAPSYS supports detector design and allows one to understand different mechanisms occurring in the analysed structures. The dark current density experimental data were compared with theoretical results at a temperature of 300 K for short, medium, and long wavelength infrared ranges. The dark current density of detectors optimized for different wavelengths was determined using various generation–recombination mechanisms. Proper matching between experimental and theoretical data was obtained by shifting the Shockley–Read–Hall carrier lifetime and the Auger–1 and Auger–7 recombination rates. Exemplary spectral responses were also discussed, giving a better insight into detector performance. The matching level was proven with a theoretical evaluation of the zero-bias dynamic resistance–area product (*R*_0_*A*) and the current responsivity of the designed detectors.

## 1. Introduction

Mercury cadmium telluride is currently the most widely used material for the production of infrared detectors [1]. It is a ternary compound (Hg_1−*x*_Cd_*x*_Te) whose great advantage is the ability to change the energy gap by changing the stoichiometric composition, *x* [2]. This allows detectors to be optimised for any wavelength in a wide spectral range. Unfortunately, HgCdTe is a technologically difficult material, and this results in its high price [2]. The vulnerability of the HgCdTe surface to environmental factors and complex difficulties related to the growth process of heterostructures, in particular the interdiffusion processes of constituent elements during epitaxial growth, are extremely challenging for device technology and engineering. Despite these limitations, the HgCdTe material is superior to other materials used in infrared detection due to its many advantages [3]. In addition to the above, wide bandgap tuning range, complete flexibility in donor and acceptor doping concentrations [4,5], high electron mobility, and low dielectric permittivity are valuable advantages. The low variability of the lattice constant with the change in compound composition facilitates the use of energy-gap engineering in the construction of devices. Thanks to this, it is possible to design complex heterostructures necessary for non-equilibrium photodiodes [6], avalanche photodiodes [7,8], and other photoelectric devices [2,6,9].

Currently, the dominant methods for obtaining HgCdTe heterostructures are epitaxial techniques such as liquid-phase epitaxy (LPE) [10], molecular-beam epitaxy (MBE) [4], and metal–organic chemical vapour deposition (MOCVD) [5]. The conducted analyses showed that the MOCVD technology is more suitable for the flexible production of detectors operating at room temperature. This is determined by such factors as higher growth efficiency, easier acceptor and donor doping at medium and high levels, and lower cost of equipment maintenance. A number of concepts to improve HgCdTe infrared detector performance and reach high-operating-temperature (HOT) conditions have been proposed [11]. For many years, the parameters of various photovoltaic detectors, especially their current–voltage (*I*–*V*) characteristics, have been studied and analysed [12,13]. However, there are only few works devoted to HgCdTe detectors operating at room temperature.

The results of research on uncooled HgCdTe detectors optimized for different spectral infrared ranges are presented. The theoretical current–voltage and spectral characteristics of these detectors were analysed and compared with experimental results. All results presented in this work, including all figures, concern the data obtained at room temperature, assumed by default to be 300 K.

## 2. Motivation and Novelty

At present, most of the designed infrared detectors require to be cooled to inhibit noises originating from thermal generation processes. Any type of cooling system increases detector size, weight, power consumption, and cost. In this context, research on detectors that do not require cooling is particularly justified. The uniqueness of the developed detectors is that they can operate at room temperature. The responsivity values of the presented HgCdTe photovoltaic detectors are comparable to the responsivity values of popular bolometers widely found in the current infrared market. However, the advantage of the presented detectors in comparison with bolometers (included in the group of thermal detectors) manifests in their response time. The time constant of bolometers is typically expressed in milliseconds, which excludes them from high-frequency applications. The time constant of the presented photovoltaic detectors is expressed in single nanoseconds, so they are approximately six orders of magnitude faster than thermal detectors.

## 3. Photodiode Design

The Hg_1−*x*_Cd_*x*_Te structures were grown in an Aixtron AIX–200 MOCVD system. Growth was carried out in 2–in, epi-ready, semi-insulating (100) GaAs substrates. A description of the MOCVD system and the growth process was presented in our previous papers [14,15]. The classical *N*^+^/*π*/*P*^+^ structure for IR detection includes a lightly doped *p*-type (*π*) semiconductor absorber with the energy gap corresponding to wavelengths in the SWIR, MWIR, and LWIR ranges. The thickness of an absorber layer is a compromise between requirements of low thermal generation and high absorption. Figure 1 demonstrates the diagram of the analysed detector structure, which consists of the following layers: lower contact layer *N*^+^ (CL), graded layer (G1), absorber layer (AL), graded layer (G2), and upper contact layer *P*^+^ (CL).

The comprehensive procedure for manufacturing detectors is extremely complex and is usually carried out by several human teams particularly responsible for theoretical modelling, design, epitaxial growth, processing, and characterization. Figure 2 shows an example of the procedure algorithm illustrating the characteristic steps in detector design, construction, and characterisation. This paper focuses on part of the whole process and mainly concerns the electro–optical characteristics of the detectors under investigation. Especially, the current–voltage theoretical and experimental data were analysed and compared (blue contours and arrows in Figure 2).

The work compares detectors designed to operate at 300 K in four spectral ranges: SWIR (*λ*_co_ = 3.3 µm), MWIR (*λ*_co_ = 4.6 µm), LWIR1 (*λ*_co_ = 6.7 µm), and LWIR2 (*λ*_co_ = 8.8 µm). A detailed description of the tested samples can be found in Table 1.

Theoretical modelling of the HgCdTe heterostructures was performed using the SimuApsys platform (Crosslight Software Inc. Vancouver, BC, Canada) [16]. The model used takes into consideration optical and electrical properties, considering radiative, Auger (Aug–1 and Aug–7), and Shockley–Read–Hall (SRH) generation–recombination mechanisms [17]. Theoretical modelling took these mechanisms into account by adding generation–recombination (G–R) coefficients *C_n_* and *C_p_* (corresponding to Auger G–R), and radiative coefficient and SRH lifetimes *τ_SRH__*_n_ and *τ_SRH_*__p_ to the macro as input parameters. Auger recombination coefficients *C_n_* and *C_p_* were determined based on the following equations [18,19]:(1)$$C_{n}=\frac{8\cdot {\left(2\pi \right)}^{2.5}\cdot q^{4}\cdot m_{0}}{h^{3}\cdot {\left({4\pi \varepsilon }_{0}\varepsilon_{s}\right)}^{2}}\cdot \frac{\left(m_{e}/m_{0}\right)\cdot {\left|F_{12}\right|}^{2}}{n_{i}^{2}\cdot {\left(1+\mu \right)}^{0.5}\cdot \left(1+2\mu \right)}\cdot {\left(\frac{k_{B}T}{E_{g}}\right)}^{1.5}\cdot e^{\left(-\;\frac{1+2\mu }{1+\mu }\cdot \frac{E_{g}}{k_{B}T}\right)}$$
(2)$$C_{p}{=C}_{n}/\gamma$$
where *q* is the electron charge, *m*_0_ is the free electron mass, *F*_12_ is the Auger overlap parameter (Bloch overlap integral), *k_B_* is the Boltzmann constant, *T* is the temperature, *E_g_* is the energy bandgap, *m_e_*/*m*_0_ is the electron effective mass (conduction band), *μ* is the electron-to-heavy-hole effective mass ratio, and *ε*_0_ and *ε_s_* are the dielectric and static frequency dielectric constants. The *γ* parameter describing the ratio of Auger–7 to Auger–1 intrinsic times is highly uncertain. Typical values are within the range of 3 < *γ* < 6 [20], and for the purpose of this paper, they were changed to the range from 5 to 15.

## 4. Experimental Results

In order to determine the basic detection parameters of the photodiodes, their spectral and current–voltage characteristics were studied. Figure 3 shows the dependence of the current responsivity (*R_i_*) as a function of the wavelength of four structures tested at 300 K. The dashed lines represent the characteristics measured without bias voltage, and the solid lines denote measurement results when the negative bias voltage was switched on. In this case, *R_i_* values could be increased by even one order of magnitude. The values of the long-wavelength cut–off were consistent with the width of the energy gap of a given absorber (Table 1). The longer the cut-off wavelength is, the lower the current responsivity of the detector at zero bias is. In order to increase it, the detector should be biased in the reverse direction. Only in the case of the SWIR detectors, due to the high dynamic resistance of the junction, the current responsivity was the same at zero voltage and reverse biasing.

Figure 4 shows the current–voltage characteristics of the measured detectors (solid lines). We can see that with the increase in the energy gap, the dark current values decreased in the reverse direction. When the energy gap of the absorber was smaller (MWIR and LWIR detectors), the Auger suppression effect on the *I*–*V* characteristics was invisible. Auger suppression in HOT HgCdTe devices translates into a unique negative differential resistance in the reverse bias *I*–*V* characteristics. Figure 4 also compares the experimental curves with the theoretical *I*–*V* characteristics determined with the typical values of the material parameters, assuming that SRH recombination did not occur and that the *F*_12_ overlapping integral in the calculation of the Auger G–R coefficients (*C_n_* and *C_p_*) was equal to 0.2. In all cases, the theoretical characteristics had lower dark currents than the experimental ones. In order to adjust them to the experiment, the material parameters of the individual layers of the detector should be corrected, and their impact on the shape of the *I*–*V* characteristics should be analysed.

## 5. Theoretical Simulation

Theoretical simulations allowed us to determine the shape of the band structure of the tested detectors. Figure 5 shows, as an example, the location of the valence band and the conduction band at a bias voltage of zero and a reverse polarization of −0.6 V for the SWIR and LWIR structures. In these figures, the position of quasi-Fermi levels is marked with a dashed line. In the case of the SWIR structure, the entire potential was deposited at the *p*–*n* junction, while in the LWIR2 structure, we observed a decrease in part of the potential at the *p*–*p*^+^ junction due to the much lower dynamic resistance of the *p*–*n* junction at 300 K.

The generation and recombination of carriers mainly occurs in the absorber region and strongly depends on the level of Auger, radiative, and SRH recombination. For example, in the SWIR structure, SRH generation predominated at the reverse bias, while the contribution of Auger recombination was negligible (Figure 6). The main influences on the carrier generation rate were the SRH lifetime and the *F*_12_ overlap integral, which strongly depends on the stoichiometric composition of the absorber. Of course, these changes had a significant impact on the shape of the *I*–*V* characteristics. Figure 7 shows how the *I*–*V* characteristics of the MWIR structure changed under the influence of SRH lifetime and the *γ* parameter.

As the SRH lifetime increased, the value of the dark current decreased, and Auger recombination suppression increased; as a result, the maximum of the dark current at reverse bias shifted towards lower voltages (Figure 7a). Figure 7b shows the dependence of the dark current as a function of voltage for three different values of parameter *γ* = *C_n_*/*C_p_*. Auger recombination suppression increased with the increase in the γ parameter, and the maximum of the dark current shifted towards lower voltages. A similar situation occurred in the LWIR2 structure, as it is shown in the contour maps in Figure 8. As SRH recombination lifetime *τ_SRH_* increased, the current density decreased. Auger recombination suppression appeared at higher values of *F*_12_ (Figure 8b) and only occurred in a narrow range of SRH lifetime values.

The dark *I*–*V* characteristics for different values of the SRH lifetime are shown in Figure 9. These characteristics were based on the dark current density distribution taken from Figure 8b. We can see how small changes in the SRH lifetime caused a significant change in the *I*–*V* characteristics. With longer SRH lifetime, Auger suppression was weaker, while at *τ_SRH_n_* = 2–3 ns, Auger suppression was strong and caused a rapid change in the dark current. This change resulted from the strong suppression of Auger generation shown in Figure 10. When the voltage changed from −0.34 V to −0.37 V, the Auger generation rate decreased by five times, and the SRH recombination rate increased and maintained a constant value for higher voltages; consequently, the dark current saturated (Figure 9).

## 6. Results and Discussion

The research results presented above show the considerable influence of the individual material parameters on the *I*–*V* characteristics. Bearing in mind these observations and selecting the appropriate values of SRH lifetime, *F*_12_, and *γ*, a good fit of the theoretical *I*–*V* characteristics to the experimental ones was obtained, as shown in Figure 11. Only with forward biasing for LWIR1 photodiodes, slightly higher theoretical dark current densities were observed in relation to the experiment. The values of fitting parameters for the individual spectral ranges are presented in Table 2. As the energy gap decreased, the carrier lifetime related to SRH recombination decreased, and overlap integral *F*_12_ increased.

The theoretical values of the photocurrent were also compared with the experimentally obtained spectral characteristics *R_i_*. The photocurrent was determined as the difference between the illuminated photocurrent and the dark current. The illuminated photocurrent was obtained when the structure was irradiated at the wavelength corresponding to the maximum current responsivity. The dependences of the photocurrent determined in this way as a function of the bias voltage of the tested structures are shown in Figure 12. The values of the photocurrent continued to increase with the increase in the cut–off wavelength at the reverse biases, while at zero voltage, we could observe the opposite tendency. These observations are consistent with the current responsivity *R_i_* characteristics determined experimentally. For the SWIR photodiodes, *R_i_* had the lowest values and was independent of the bias voltage, while for the LWIR photodiodes, *R_i_* had the highest values at negative voltage, and it strongly dropped at zero bias (see Figure 3). This decrease was caused by a low value of the dynamic resistance of the junction at room temperature, practically comparable to the series resistance of the LWIR detector.

Finally, we compared the experimental values of the *R*_0_*A* product with its theoretical limits as a function of absorber composition *x* (Figure 13). The theoretical data are only based on Auger and radiative recombination. For all spectral ranges discussed, our experimental values are more than twice as low as the theoretical limits, so there is a possibility of further improvement of the constructed detectors. The *R*_0_*A* products for the LWIR detectors are the closest ones to their theoretical limits. Figure 13 also shows the *R*_0_*A* product experimental data taken from the literature [1] (see page 652).

## 7. Conclusions

Appropriate adjustment between the experimental and theoretical data of the HOT HgCdTe detectors was obtained by fitting the lifetime of carriers for Shockley–Read–Hall recombination and Auger–1 and Auger–7 recombination rates. The conducted analysis showed that the current–voltage and spectral characteristics of heterojunction photodiodes operating at 300 K depend strongly on the stoichiometric composition of the absorber. It was confirmed that photodiodes in the SWIR range are characterised by the longest lifetime for SRH G–R processes. In the case of photodiodes in the LWIR spectral range, Auger recombination suppression plays a dominant role, which, at the same time, strongly depends on the value of carrier lifetime determined by SRH recombination. Hence, the mutual matching of LWIR characteristics is more difficult. Finding the relationship between the theoretical and experimental characteristics allows appropriate corrections to be introduced in the modelling formulas and, as a consequence, results in a better understanding of the nature of HOT HgCdTe infrared detectors.

## Figures and Tables

**Figure 1 sensors-23-01088-f001:**
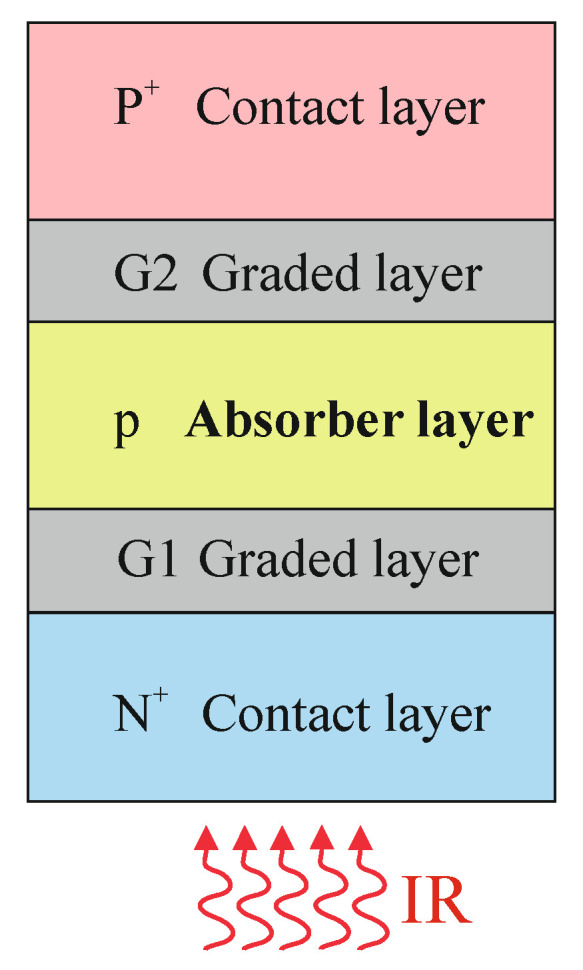
General scheme of the analysed detector structure.

**Figure 2 sensors-23-01088-f002:**
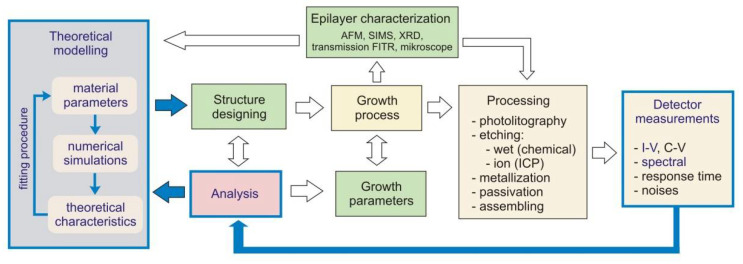
Typical procedure algorithm illustrating the characteristic steps in detector design, construction, and characterisation.

**Figure 3 sensors-23-01088-f003:**
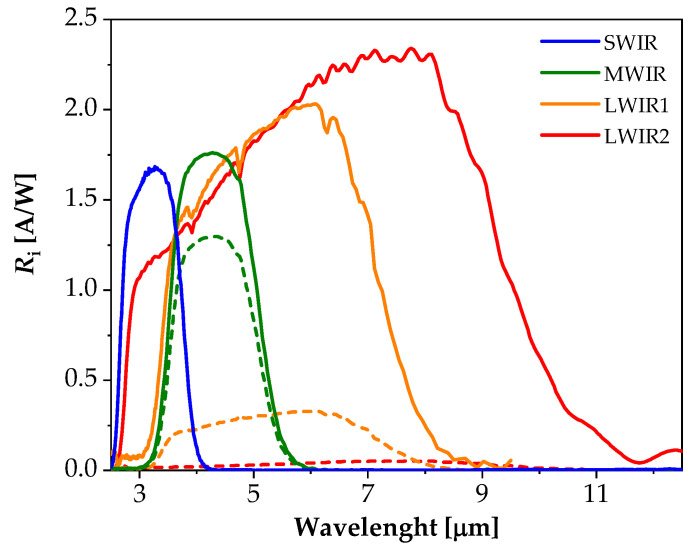
Spectral characteristics of SWIR, MWIR, and LWIR diodes without (dashed line) and with (solid line) reverse bias voltage measured at a temperature of 300 K.

**Figure 4 sensors-23-01088-f004:**
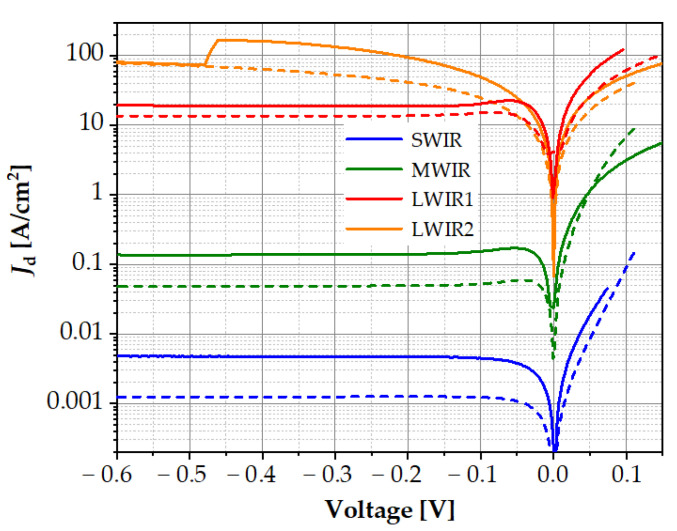
Experimental (solid line) and theoretical (dashed line) *I*–*V* characteristics of SWIR, MWIR, and LWIR diodes at the temperature of 300 K.

**Figure 5 sensors-23-01088-f005:**
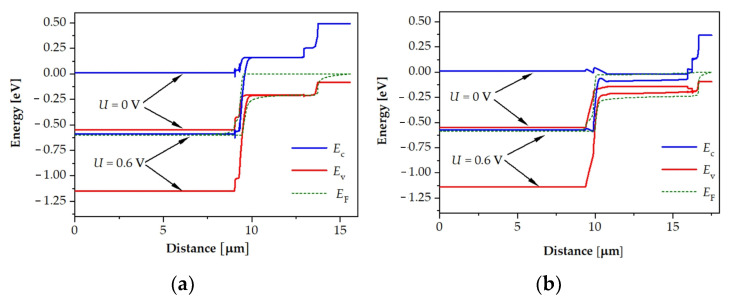
Energy band diagram for the SWIR (**a**) and LWIR2 (**b**) structures with and without bias voltage at temperature T = 300 K.

**Figure 6 sensors-23-01088-f006:**
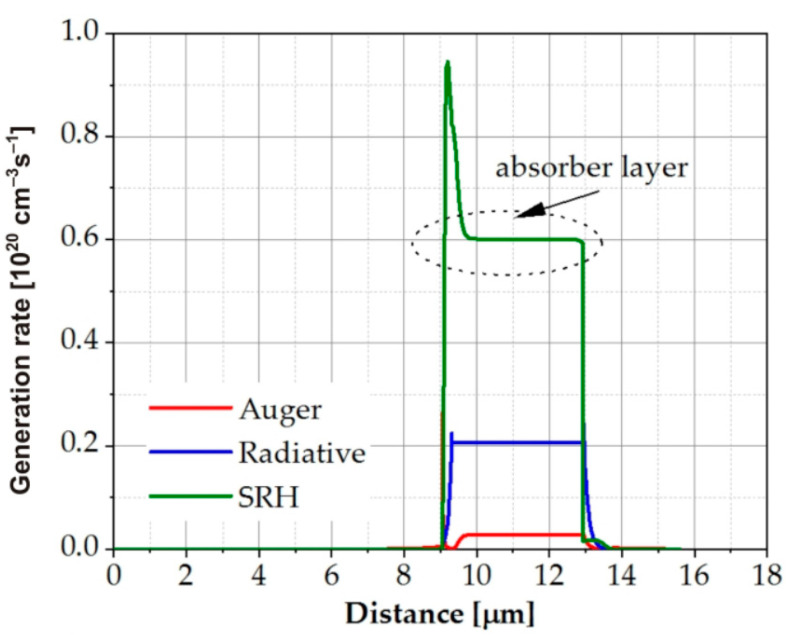
Auger, radiative, and SRH generation rates in the reverse biased SWIR detectors at temperature T = 300 K.

**Figure 7 sensors-23-01088-f007:**
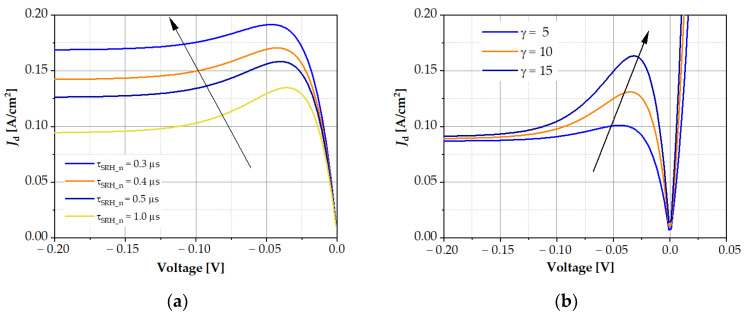
Current–voltage characteristics of the modelled MWIR detector structure: (**a**) for different SRH electron recombination times and (**b**) for different Auger *γ* parameters.

**Figure 8 sensors-23-01088-f008:**
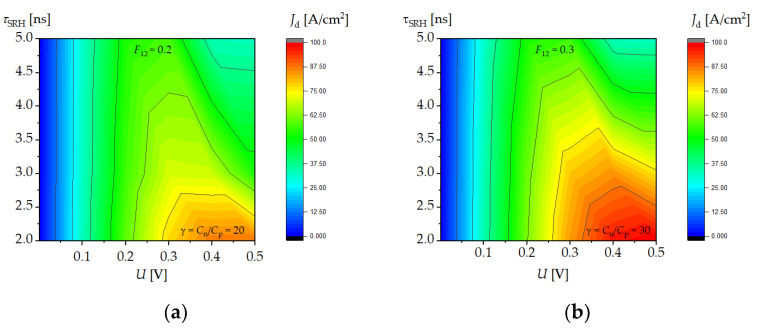
Distributions of the dark current density in the reverse direction as a function of voltage and SRH lifetime in the LWIR2 structure for (**a**) *γ* = 20, *F*_12_ = 0.2; and (**b**) *γ* = 30, *F*_12_ = 0.3.

**Figure 9 sensors-23-01088-f009:**
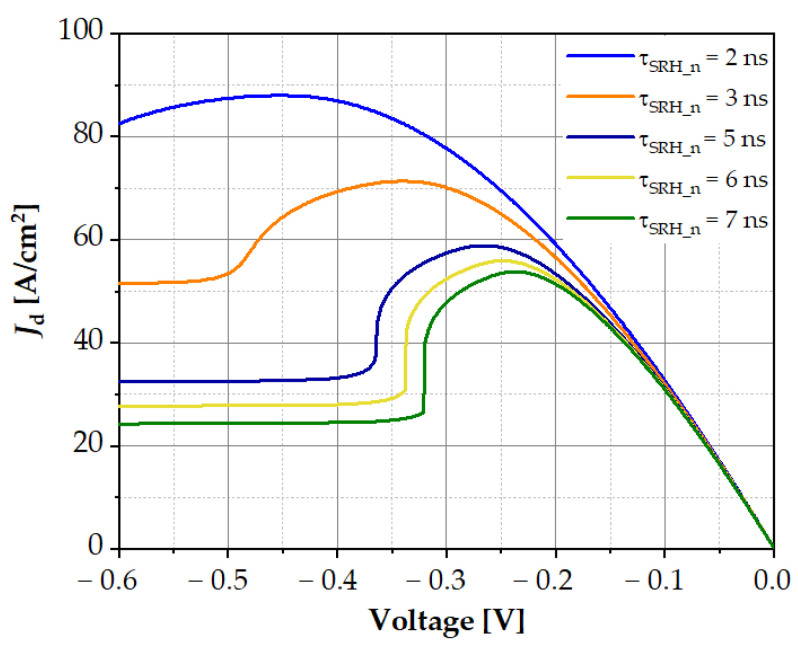
Calculated dark current versus voltage of the LWIR2 structure for different SRH recombination lifetimes at temperature T = 300 K.

**Figure 10 sensors-23-01088-f010:**
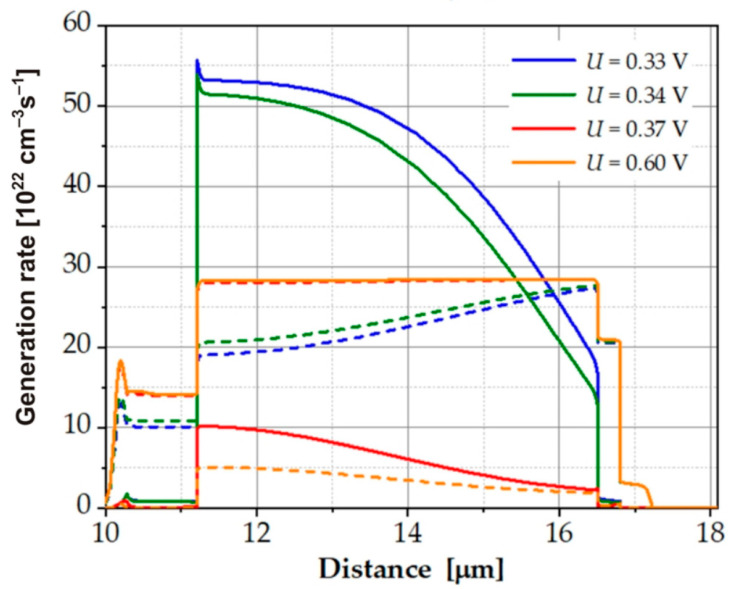
Auger (solid line) and SRH (dashed line) generation rates in reverse bias for the LWIR2 detector at T = 300 K.

**Figure 11 sensors-23-01088-f011:**
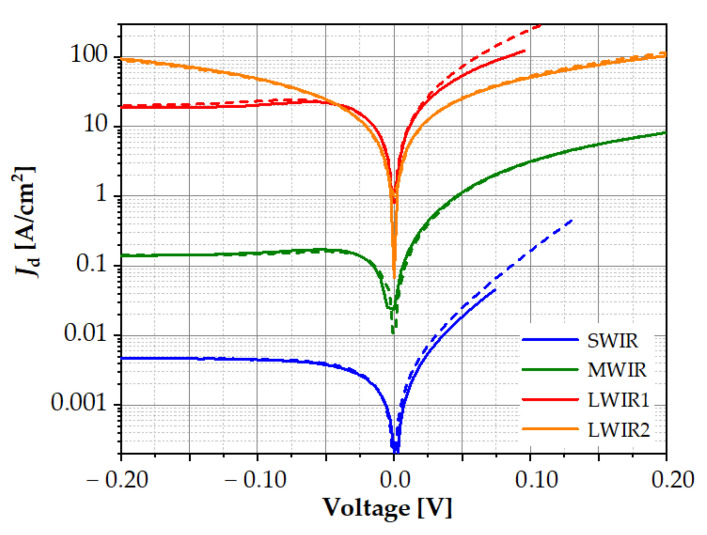
Current–voltage characteristics of the SWIR, MWIR, and LWIR diodes taken at 300 K. The solid and dashed lines denote experimental and theoretical data, respectively.

**Figure 12 sensors-23-01088-f012:**
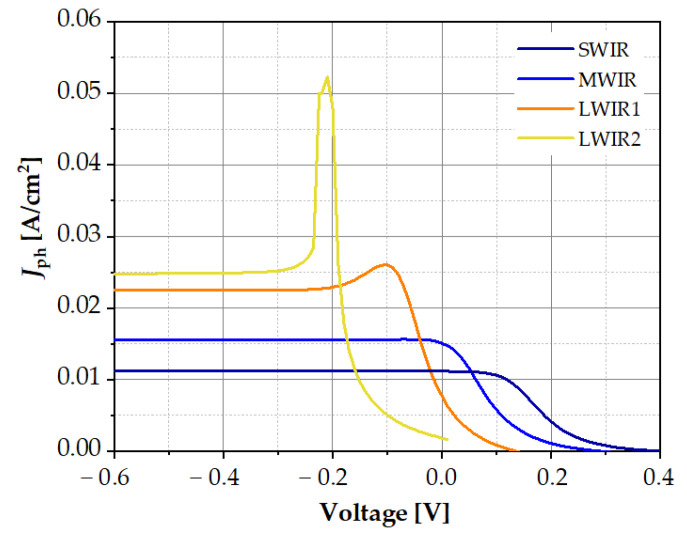
Photocurrent versus voltage for measured structures of different spectral ranges at T = 300 K.

**Figure 13 sensors-23-01088-f013:**
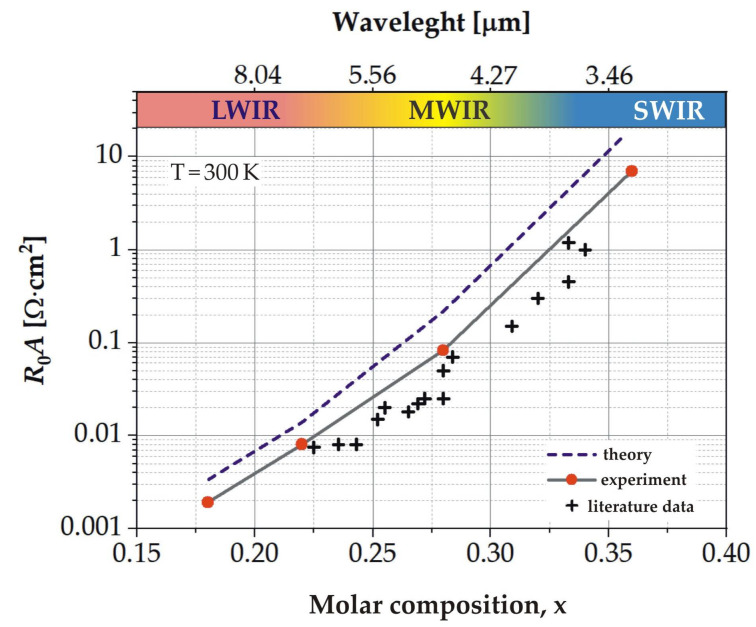
Theoretical and experimental data of the *R*_0_*A* product as a function of molar Cd composition *x* at room temperature. Literature data are from [1].

**Table 1 sensors-23-01088-t001:** Thickness, molar composition, and dopant concentration of individual layers of the SWIR, MWIR, and LWIR detector structures.

Layer	Thickness, *d* (µm)				Composition, *x*				*N*_A_-*N*_D_ (cm^−3^)
	SWIR	MWIR	LWIR1	LWIR2	SWIR	MWIR	LWIR1	LWIR2	
*P* ^+^	0.8 ÷ 2.0				0.3 ÷ 0.5				+2 ÷ 5 × 10^17^
G2	0.4 ÷ 0.8				0.25 ÷ 0.42				+5 ÷ 7 × 10^15^
*p*	3.0	4.0	5.1	5.2	0.36	0.28	0.22	0.18	+3 ÷ 7 × 10^15^
G1	0.3	0.3	1.5	1.5	0.42	0.36	0.36	0.25	−1.0 × 10^17^
*N* ^+^	9.0				0.4 ÷ 0.5				−2.0 × 10^17^

**Table 2 sensors-23-01088-t002:** HgCdTe material parameters at 300 K obtained after fitting simulated and measured *I*–*V* characteristics for different spectral ranges.

Parameter	SWIR	MWIR	LWIR1	LWIR2
*F* _12_	0.1	0.255	0.25	0.2
γ	5	8	10	5
*τ_SRH_n_* (ns)	1400	400	15	0.3

## Data Availability

The data presented in this study are available on request from the corresponding author.

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
