# Peer review of "Research on Electro-Optical Characteristics of Infrared Detectors with HgCdTe Operating at Room Temperature"

_sensors, 2023, doi:10.3390/s23031088_

Round 1

Reviewer 1 Report

Summary comment: The overall intention of the paper “Researches on electro-optical characteristics of infrared detectors with HgCdTe operating at room temperature” by PaweÅ‚ Madejczyk et al., is a good one. Because of their unique electrical and optical properties and their potential applications in infrared photodetection, I agree that more attention should be paid to investigate the correlation between the electronic and optical properties of compounds materials, such as HgCdTe. These attempts aide us to improve the MCT based photodetectors performance operating at RT.

After a careful reading of this submission these are my remarks for the Editor/Author to consider:

Comment 1:

- Table 1: Check the values of d(p) and d(G1)

- Line 94: me/m0 is the electron effective mass.

-   Line 109-110: can you justify in the case of SWIR detectors, why the current responsivity is the same at zero voltage and reverse biasing.

- Figure 3: in the case of LWIR1, where is the theoretical I-V characteristic?

-  Line 125: (soli line)>> (solid line), (dash line) >>> (dashed line)

-  Lines 145-147: Figure 6 do not show the effect of the Bloch function overlap integral parameter F12 on I-V characteristics?

- Line 161: 3D plot >>> contour maps

- Line 185: The considerations led in the previous chapter??

- Figure 12: The authors are invited to show all the discussed spectral ranges.

-   Line 223: 4. Conclusion >>> 6. Conclusion

- Line 231: plays an dominant role>> plays a dominant role,

-  …………….

Comment 2The authors should work on improving the fluency and grammatical part in the entire manuscript.

Comment 3: The introduction section should be improve, updated and extended with studying and citation of new papers.

Comment 4: The author should include a brief motivation and novelty section in his manuscript to explain the significance of his intervention in the field.

Comment 5: In the experimental section, the authors can use a schematic illustration for the experimental procedure used to characterize HgCdTe alloy for the study.

Comment 6: The physical explanations of the crucial findings in the entire of this manuscript require an improvement.

Comment 7: The authors are advised to clarify the correlation between electrical and optical properties of the investigated MCT based photodiodes.

Comment 8: I will be glad if the authors validate their results by the available findings in the literature?

Comment 9: Some references that are more than 20 years old should be replaced with more recent ones.

Comment 10: Pertaining to environmental influences how much HgCdTe will have an impact on the environment?

Reviewer 2 Report

This article proposes a thorough analysis of the current-voltage characteristics of uncooled HgCdTe detectors optimized for different spectral ranges. The obtained results are interesting, but the paper cannot be published in the current version. There are several suggestions:

1 This article studies the newly designed photodiode, but there is no comparison experiment, please add it.

2 In part 3, The author gives a lot of experimental results, but there is no analysis of the results.

3 The author has a lot of work, but the purpose and significance of the experiments are not expressed in the article and need to be supplemented.

5. The references are relatively out-of-date, some new papers within three years should be cited.

6. There are some grammatical errors, and the author should tackle the problems of the English tense.

Round 2

Reviewer 1 Report

The Submission has been improved and the revision is clearly presented. Especially, Figure 2 is very helpful to understand the general procedure, with different steps, used in the photodetector design. Thereby, I think most of my suggestions and comments have been well addressed with either necessary discussion.

There are still some minor revisions for the authors to consider before consideration.

Line 17 in the Abstract: Different physical phenomena determine the dark current density for each spectral range??>>> To rephrase

Line 31-34: , in particular to interdiffusion processes of constituent elements during epitaxial growth 33 which are extremally challenging for the devices' technology and engineering >>>, in particular the interdiffusion processes of constituent elements during epitaxial growth, are extremely challenging for the devices' technology and engineering

Line 37: In addition to the above-mentioned, >>> In addition to the above-mentioned;

Line 48: near room temperature>>> room temperature

Line 56: The paper presents >>>

Line 64: Sensors>>> detectors

Comment: The authors are invited to remove the background of figure 2.

Author Response

Line 17 in the Abstract: Different physical phenomena determine the dark current density for each spectral range??>>> To rephrase

The sentence was rephrased into form:

“The dark current density of detectors optimized for different wavelengths is determined by various generation-recombination mechanisms.”

Line 31-34: , in particular to interdiffusion processes of constituent elements during epitaxial growth 33 which are extremally challenging for the devices' technology and engineering >>>, in particular the interdiffusion processes of constituent elements during epitaxial growth, are extremely challenging for the devices' technology and engineering

The sentence was amended according to Reviewer suggestion:

Vulnerability of the HgCdTe surface to environmental factors and complex difficulties related to the growth process of heterostructures, in particular the interdiffusion processes of constituent elements during epitaxial growth, are extremely challenging for the devices' technology and engineering.

Line 37: In addition to the above-mentioned, >>> In addition to the above-mentioned;

The phrase was amended into the form: “In addition to the above”

Line 48: near room temperature>>> room temperature

Ok, corrected.

Line 56: The paper presents >>>

The sentence was rephrased into form: “The results of research on uncooled HgCdTe detectors optimized for different spectral infrared ranges are presented.”

Line 64: Sensors>>> detectors

Ok, corrected.

Comment: The authors are invited to remove the background of figure 2.

The background was removed in figure 2.

Reviewer 2 Report

No more comments

Author Response

Minor English edits were corrected.